# U-Model-Based Two-Degree-of-Freedom Internal Model Control of Nonlinear Dynamic Systems

**DOI:** 10.3390/e23020169

**Published:** 2021-01-29

**Authors:** Ruobing Li, Quanmin Zhu, Pritesh Narayan, Alex Yue, Yufeng Yao, Mingcong Deng

**Affiliations:** 1Department of Engineering Design and Mathematics, University of the West of England, Frenchay Campus, Coldharbour Lane, Bristol BS16 1QY, UK; quan.zhu@uwe.ac.uk (Q.Z.); Pritesh.Narayan@uwe.ac.uk (P.N.); Alex.Yue@uwe.ac.uk (A.Y.); Yufeng.Yao@uwe.ac.uk (Y.Y.); 2Department of Electrical and Electronic Engineering, Tokyo University of Agriculture and Technology, 2-24-16 Nakacho, Koganei, Tokyo 183-8538, Japan; deng@cc.tuat.ac.jp

**Keywords:** Internal Model Control (IMC), U-model, U-model-based control (U-control), Two-Degree-of-Freedom IMC (TDF-IMC), dynamic inversion, invariance entropy

## Abstract

This paper proposes a U-Model-Based Two-Degree-of-Freedom Internal Model Control (UTDF-IMC) structure with strength in nonlinear dynamic inversion, and separation of tracking design and robustness design. This approach can effectively accommodate modeling error and disturbance while removing those widely used linearization techniques for nonlinear plants/processes. To assure the expansion and applications, it analyses the key properties associated with the UTDF-IMC. For initial benchmark testing, computational experiments are conducted using MATLAB/Simulink for two mismatched linear and nonlinear plants. Further tests consider an industrial system, in which the IMC of a Permanent Magnet Synchronous Motor (PMSM) is simulated to demonstrate the effectiveness of the design procedure for potential industrial applications.

## 1. Introduction

With the development of science and technology, the scale of industrial production in almost all fields, such as petrochemical, metallurgy, electric power, machinery, aerospace, etc. continues to expand, and the corresponding operational systems have had a demand for high quality and better quantity [1], which is inevitably at the cost of bringing complexity to the control system design. These challenges have motivated academic research and industrial development.

The classical Proportional Integral Derivative (PID) control and its integrations with various control strategies such as fuzzy PID [2], and neural PID [3] have been widely used in industrial systems. Although these control strategies can cope with complexities such as uncertainty, nonlinear dynamics, and large time delays, it is still worthwhile seeking other effective control system design methodologies to further upgrade control system performance while improving the design effectiveness. For example, a commonly observed practical situation, is that the success of tuning PID controller parameters often depends on a combination of the applicant’s engineering experience and tedious effort on trial and error. Although this is workable, this unsystematically experienced approach often causes largely inefficient use of human resource and equipment to obtain satisfactory tuning results.

For this paper’s interest, model validity is a fundamental basis for model-based control system design. A better model makes control system design and tuning easier/more efficient. However, for most engineering systems, there can be difficulties in obtaining accurate plant/process models, primarily due to equipment diversity and environment complexity; such as internal uncertainties and external disturbances. Even though a mathematical model can be established from physical principles (such as energy conservation law) and/or data driven (identification), it is usually taken as nominal reference (nominal model is an approximate description to accurate model). There are two main streams in dealing with such model uncertainties, i.e., adaptive control and robust control. This paper will follow the route of robust control.

The other practical topic is digital control, which has been used in almost all modern control engineering systems. To deal with a digital channel between the sensor and the controller, entropy [4], the concepts, terminologies, and techniques, has been adopted in digital control systems. The invariance entropy [5] has been used to determine the smallest average rate of information transmission to guarantee a considered subset of the state-space invariant to achieve the integrated control system performance. Accordingly control system design in connection to online applications, with potential contribution to this field, is probably making the controllers more efficient in online computation, so that reduces the burden to the communication capacity or at least does not increase significantly the information processing complexity.

Internal Model Control (IMC) [6,7] has been widely accepted as an efficient robust control approach. IMC selects the model inverse as the controller and integrates a robust filter to control an explicit plant/process model. The IMC structure is characterized with (1) capable robustness to overcome model uncertainties and system disturbances, (2) effective procedures for designing and tuning, (3) successful application across different industries [8,9,10]. However, the control performance of classical IMC is not desirable, because the adjustable parameters only exist in the filter. At the same time, higher robustness demand could degrade tracking performance [11], which must compromise with some of the other performances. Although a Two-Degree-of-Freedom IMC (TDF-IMC) structure can solve the aforementioned problems with the classical IMC structure, its control performance still cannot be separately designed [12,13,14].

When a linear model is completely reversible, the design of linear IMC is straightforward to take the controller as the inverse of the model and select a suitable filter. Even though when the model is not completely reversible, the model can be decomposed into reversible parts and irreversible parts, in which the inverse of the reversible part is taken as the controller. Appropriate filter selection can then also ensure that the control system has the smallest output variance for both stabilization and tracking control. However, for controlling nonlinear plants/processes, these approaches are not applicable, and effective algorithms for nonlinear dynamic inversion are very limited [15].

To deal with nonlinear control plants/processes, the approaches used by most of the IMC structures can be divided into (1) linearizing the controlled plants/processes and using linear method to invert [16]; (2) using PID [17], neural network [18,19] and fuzzy control-based [20,21] dynamic inversion; (3) using some numerical tools, such as the Newton–Raphson method [22]. However, the linearized and the other approximating modeling methods could lose accurate representation of input-output relationship and degrade the performance of the designed systems. Therefore, deriving the nonlinear model inversion and enabling the two performance indicators (i.e., tracking and robustness) of the IMC structure to be independently designed are the main challenges and focuses in this paper. Accordingly, this study proposes a framework of U-Model-Based Two-Degree-of-Freedom Internal Model Control (U-TDF-IMC) of nonlinear dynamic systems

U-model is a derived control-oriented model set to map almost all classical models into their U-model realization, and converts classical models into controller output u-based with time-varying parameters [23] expressions. U-model establishes a platform for solution of dynamic inversion by solving roots of polynomial equations, which is more generally attractive compared to the other ad hoc approaches/algorithms [24]. U-model-based control [25] (denoted as ‘U-control’ thereafter), takes advantages of U-model in dynamic inversion with the following characteristics:
Design control systems in a universal procedure, separate two dynamic inversions, invariant controller implementation by inversing specified system performance in a feedback configuration and plant utilization by plant inversion. These two designs are parallel and separable;The difference in U-control between linear models and nonlinear models is the solution with the first-order or higher-order polynomial root-solving. The difference in U-control between polynomial models and state-space models is the one-layer or multi-layer polynomial root-solving;U-control is seamlessly supplemented to the other exist control schemes, for example U-Pole Placement Control (UPPC) [26], U-General Predictive Control (UGPC) [27], U-Neuro-Control (UNC) [28], U-Total Nonlinear Control (UNLC) [23], and U-Internal Model Control (UIMC) [29].

This paper is aimed at using U-control, an enhanced tool supplemented to classical approaches, to integrate the strengths exhibited in U-control and IMC to provide an enhanced version of IMC with strength in system configuration and nonlinear dynamic inversion. To further improve TDF-IMC, this study expands the previous IMC work [29,30] by effectively introducing U-model-based dynamic inversion within a revised system structure configuration. By doing so, the new framework presents a new U-model-based Two-Degree-of-Freedom IMC (UTDF-IMC) structure to achieve the completely independent design in rejecting disturbance and tracking operational set-point. Compared with the classical IMC and TDF-IMC, this proposed structure has better control performance and more convenient tuning methods without introducing additional design work and maintaining the same hardware configuration. Accordingly, the major impacts of this paper are outlined below:Propose a general U-model-based Two-Degree-of-Freedom IMC (UTDF-IMC) structure for controlling a class of open-loop stable polynomial/state-space modeled linear and nonlinear dynamic plants. The new control system structure accommodates both linear and nonlinear plants consistently and separate the tracking control filter design from robust control filter design.Tailor the UM-dynamic inversion platform [31] in conjunction with IMC, which removes the necessity of either linearizing the nonlinear model, or converting it to a quasi-linear parameter-varying (quasi-LPV) model in advance. This UM-dynamic inversion platform directly provides algorithms dealing with all types of inversions in IMC structured systems.Analyze the designed UTDF-IMC control system properties to provide a valid reference for future study expansions and applications.Verify the control system performance through benchmark tests of simulated case studies and illustrate application procedure from an industrial case demonstration.

For the remainder of the paper, Section 2 presents the basis of using IMC and U-control for the next step development of the new UTDF-IMC system structure. Section 3 elaborates on the principle of TDF-IMC structure and establishes the U-model-based TDF-IMC (UTDF-IMC) framework; consequently, it analyses the control system properties. Section 4 showcases two computational investigations to benchmark test/demonstrate the proposed UTDF-IMC system performance. Then an industrial backgrounded permanent magnet synchronous motors (PMSM) system is added to demonstrate the application procedure and the comparative studies. Section 5 concludes this study with key findings and observations.

## 2. Preliminaries

### 2.1. Internal Model Control (IMC)

A classical IMC control scheme [7] is shown in Figure 1a, in which the plants/process P is approximated by model $P_{0}$ (specifically known as internal model) and the controller Q. Figure 1b shows the equivalently rearranged IMC structure, which the controller is expressed in the inner loop
(1)$$C=\frac{Q}{1-QP_{0}}$$

For a given set-point reference r, the control system is designed to keep the output y following a pre-specified output response $y_{m}$ (Figure 1a) with the desired transient and steady-state performance. With reference to Figure 1a,

Plant output:(2)$$y=\frac{QP}{1+Q\left(P-P_{0}\right)}r+\frac{1-QP_{0}}{1+Q\left(P-P_{0}\right)}d$$

Error output:(3)$$e=\frac{1}{1+Q\left(P-P_{0}\right)}\left(r-d\right)$$

Controller output:(4)$$u=\frac{Q}{1+Q\left(P-P_{0}\right)}\left(r-d\right)$$

**Remark 1.** *(2) can also be rewritten as:*(5)y=αr+βd*where*$\alpha =\frac{QP}{1+Q\left(P-P_{0}\right)}$*specifies tracking performance and*$\beta =\frac{1-QP_{0}}{1+Q\left(P-P_{0}\right)}$*denotes the contribution to robustness. These two weights meet the condition of*α+β=1.

The main features of IMC [7] include:
Dual stability: For $P=P_{0}$ and d=0, and $y=y_{m}$, the feedback error signal e is obviously zero. IMC system becomes an open-loop structure and both controller Q and plant P stable.Perfect control: This requests plant $P=P_{0}$ minimum-phase and invertible and controller as the model inverse $Q=P_{0}{}^{-1}$. Accordingly (2) becomes:(6)$$y=\frac{P_{0}{}^{-1}P}{1+P_{0}{}^{-1}P-P_{0}{}^{-1}P_{0}}r+\frac{1-P_{0}{}^{-1}P_{0}}{1+P_{0}{}^{-1}\left(P-P_{0}\right)}d=r,\;\mathrm{and}\;\alpha =1,\beta =0$$Augmented robust IMC is shown in Figure 2. It decomposes model and dynamic inversion by factorizing $P_{0}$ into $P_{0+}$ and $P_{0-}$, namely: $P_{0}=P_{0+}P_{0-}$, where $P_{0+}$ is the part containing pure delay and uncertain zero, and $P_{0-}$ is the minimum-phase part. There are certain factorization techniques, such as simple factorization, all-pass factorization [32]. Hence, the controller is kept as the inverse of the plant/process model with invertible portion, i.e.,
(7)$$Q_{1}=\frac{1}{P_{0-}}$$

Filter: When designing the IMC controller, it should add a low-pass filter for the inverse of the factorized minimum-phase model to ensure the controller be proper and robust to against the model mismatching and disturbance. Define the IMC controller and the filter as:(8)$$Q=fQ_{1}$$
(9)$$f=\frac{1}{{\left(1+\lambda s\right)}^{\rho }}$$
where ρ is the order of the filter, normally assigned with a large value to ensure $Q_{1}$ be proper or semi-proper; λ is the time constant, the sole design parameter of the controller and is inversely proportional to the closed-loop response speed. Therefore, λ is a trade-off between the performances.

**Remark 2.** 
*Substituting (7) and (8) into (2) obtains the plant output:*
(10)$$y=\frac{\frac{f}{P_{0-}}P}{1+\frac{f}{P_{0-}}P-\frac{f}{P_{0-}}P_{0}}r+\frac{1-\frac{f}{P_{0-}}P_{0}}{1+\frac{f}{P_{0-}}P-\frac{f}{P_{0-}}P_{0}}d=\frac{\frac{f}{P_{0-}}P}{\left(1-fP_{0+}\right)+\frac{f}{P_{0-}}P}r+\frac{1-fP_{0+}}{\left(1-fP_{0+}\right)+\frac{f}{P_{0-}}P}d$$


To track the reference signal with a faster speed and effectively reject the modeling errors and system disturbance, it requires output (10) satisfying $fP_{0+}=1$ necessarily, which is achieved by selecting λ in the filter.

### 2.2. U-Model-Based Control (U-Control)

#### 2.2.1. U-Models

A general U-polynomial-model of GP [33], with a triplet of yt,ut,αt, yt∈ℝ, ut∈ℝ
αt∈ℝJ for the output, input, and time-varying parameter vector respectively at time t∈ℝ+, is defined for Single-Input and Single-Output (SISO) dynamic processes as
(11)My=ATU=∑j=0JαjfjNuj, M≥N
where My and Nu are the Mth and Nth order derivatives of the plant output y and the plant input u respectively. The time-varying parameter αj∈ℝ+ absorbing all other output terms such as m−1y,…,y∈ℝM and input terms n−1u,…,u∈ℝN. Function fj∗ is associated with the input Nuj. AT=α0,…,αJ and U=f0,…,fJ are the operators mapping the underlying input, output, and parameters into the condensed vector expressions. To illustrate the U-representation of classical models, consider a general polynomial model:(12)y¨=1−e−yy˙+1+y2sinu+1+y˙2u2+y+y+y˙2u3

Its U-model is transformed with the U-mappings of A and U
(13)$$\left\{\begin{array}{c} \ddot{y}=\alpha_{0}f_{0}\left(u\right)+\alpha_{1}f_{1}\left(u\right)+\alpha_{2}f_{2}{\left(u\right)}^{2}+\alpha_{3}f_{3}{\left(u\right)}^{3}\\ \alpha_{0}=\left(1-e^{-\left|y\right|}\right)\dot{y}+y,\;f_{0}\left(u^{0}\right)=1\\ \alpha_{1}=1+y^{2},\;f_{1}\left(u\right)=\sin\left(u\right)\\ \alpha_{2}=1+{\dot{y}}^{2},\;f_{2}\left(u^{2}\right)=u^{2}\\ \alpha_{3}=y+{\dot{y}}^{2},\;f_{3}\left(u^{3}\right)=u^{3} \end{array}\right.$$

**Remark 3.** *For representation to classical linear models, assign degree*J=1*and function*$f_{0}\left(u\right)=1,\;f_{1}\left(u\right)=u$, *then the linear U-model is expressed with*(14)$$\begin{array}{c} \left(M\right)\\ y \end{array}=\alpha_{0}+\alpha_{1}\begin{array}{c} \left(N\right)\\ u \end{array},\;M\geq N$$

**Remark 4.** 
*For U-stats space-models, expand the single layer U-polynomial model (11) into multi-layer systems of polynomials [31].*


#### 2.2.2. UM-Dynamic Inversion

For determining the output *u* of $G_{p}^{-1}$, a general UM-dynamic inversion algorithm is developed [19] to determine one of the solutions of $\begin{array}{c} \left(N\right)\\ u \end{array}$ from solving the following general polynomial equation
(15)$$G_{P}^{-1}\Leftrightarrow \begin{array}{c} \left(N\right)\\ u \end{array}\in \begin{array}{c} \left(M\right)\\ y \end{array}-\overset{J}{\underset{j=0}{\sum }}\alpha_{j}f_{j}\left(\begin{array}{c} {\left(N\right)}^{j}\\ u \end{array}\right)=0,\;M\geq N$$

For the solution exist, the systems must be Bounded Input and Bounded Output (BIBO) stable and no unstable zero dynamic (nonminimum phase). The solution platform has been expanded including the root-solving algorithms for continuous/discrete time, linear/nonlinear, polynomial/state-space models [31]. For online root-solving, Zhu [34] has proposed Newton–Raphson iterative algorithm.

#### 2.2.3. U-Control

Let Gp be a general representation of both polynomial and state-space-based linear and nonlinear models for dynamic plants. In assumption, the plant has most properties as those requested in the other classical works [35]. Consequently,
Model of Gp is invertible, i.e., GP−1 existsMeet the continuity of Lipschitz, Gp and its inverse GP−1 are globally unified and diffeomorphic in ℝn, i.e.,
(16)‖Gpx1−Px2‖≤γ1Gp‖x1−x2‖, ∀x1,x2∈ℝn‖GP−1x1−GP−1x2‖≤γ2 GP−1‖x1−x2‖, ∀x1,x2∈ℝn
where x1,x2 are the states while Gp in the expression of state-space equation, γ1 and γ2 are the Lipschitz coefficients. This study takes SISO (input u∈ℝ1 and output y∈ℝ1) prototype in consideration. U-control system framework [25] is shown in Figure 3, in which F is for U-control system structure, Gc1 is a linear invariant controller to be designed, and GP−1 is the inverter of the controlled plant GP to be designed as well. It is noted that U-control framework is applicable to various plants/processes when the dynamic inverse GP−1 exist.

The U-control system is structured
(17)$$\sum =\left(F,C\left(G_{c1},G_{P}^{-1}\right),G_{p}\right)\underset{}{\Leftrightarrow }\sum =\left(F,G_{c1},G_{ip}\right)$$
where $C\left(*\right)$ is a set to be designed and $G_{ip}=G_{P}^{-1}G_{P}$.

In general, the design of U-control system can be divided into two separate designs:Designed dynamic inverter $G_{P}^{-1}$ to achieve $G_{P}^{-1}G_{P}=1$. This gives $\sum =\left(F,G_{c1}\right)$Design invariant controller $G_{c1}$, which is a typically linear controller. Let the specified closed-loop performance in transfer function G, in form of $G=\frac{G_{c1}}{1+G_{c1}}$, which can be comfortably assigned using damping ratio ζ and undamped natural frequency $\omega_{n}$ for linear system dynamic/steady-state response.

## 3. U-Model-Based Two-Degree-of-Freedom IMC (UTDF-IMC)

### 3.1. Classical Two-Degree-of-Freedom IMC (TDF-IMC) Structure

Figure 4 shows a TDF-IMC structure to be incorporated with U-control, which comprises feedback controller F added in the external loop within the classical IMC structure. Clearly, if the feedback filter F is a unit constant, this structure is the same as that in Figure 1a.

From Figure 4, the system output $y=y_{m}+y_{e}$. Therefore,
(18)$$y=\left(r-y_{e}F\right)QP_{0}+y_{e}=rQP_{0}+y_{e}\left(1-FQP_{0}\right)$$

In the TDF-IMC system, if the controlled plant is a minimum-phase system, then the controller $Q\left(s\right)=f\left(s\right)/P_{0}\left(s\right)$. The output of (18) can be re-organised as:(19)$$y=rf+y_{e}\left(1-Ff\right)$$

The explicit input/output relationships from Figure 4 can be written as follows:(20)$$u=\frac{Q}{1+\left(P-P_{0}\right)FQ}\left(r-dF\right)$$
(21)$$y=\frac{QP}{1+\left(P-P_{0}\right)FQ}r+\frac{1-QFP_{0}}{1+\left(P-P_{0}\right)FQ}d$$

If the controlled system does not contain uncertain parameters or control disturbance, then $y_{e}=0$, otherwise, $\left|y_{e}\right|>0$. From (19), rf determines the system tracking performance, while $y_{e}\left(1-F\right)f$ determines the system robustness.

To achieve desired control performance, a condition must hold true below:(22)$$\underset{t\to \infty }{\lim}f\left(t\right)=1,\;\underset{t\to \infty }{\lim}L^{-1}\left(F\left(s\right)f\left(s\right)\right)=1$$
where $L^{-1}\left(*\right)$ is the inverse Laplace transform operator, $F\left(s\right)$ and $f\left(s\right)$ are the Laplace functions of filters F and f respectively. Thus, output y equals to the reference r eventually, and the system disturbance and modeling errors will be eliminated. The performance of the IMC control system will depend on these two filters F and f. The setting time and rise time of these two filters should be as short as possible. However, response speed which are too fast will cause the amplitude of the controller output signal to increase sharply.

From Figure 4, the controller $Q\left(s\right)$ output u is:(23)$$u=\left(r-y_{e}F\right)Q$$

From (20), when controller Q is determined, the faster the response of the filter F, the larger value the initial controller output u. In general, this can be observed from (19) that the tracking ability and robustness of IMC system cannot be separately designed, as well as its design flexibility is relatively limited. Therefore, this is one of paper aims, to separate IMC’s designing of tracking ability control and robustness and improve its design effectiveness without affecting its desired control performance.

### 3.2. U-Model-Based Two-Degree-of-Freedom IMC (UTDF-IMC) Structure

Based on the IMC problem stated in introduction and TDF-IMC analysis in Section 3.1, this paper changes the classical TDF-IMC structure in Figure 4 to a UTDF-IMC structure as shown in Figure 5.

In Figure 5, the original controller Q in classical TDF-IMC shown in Figure 4 has been split into two parts: the feedforward filter f and the inversion $P_{u}{}^{-1}$ of the U-realization controlled plant model $P_{u}$, where the original IMC’s controller $Q=fP_{0}{}^{-1}$. In contrast to the classical IMC structure, feedforward filter f appears outside the system feedback loop. However, generally the plant model inversion $P_{0}{}^{-1}$ cannot exist alone because of its irrationality and unrealizable property. For polynomial-based modeling of the controlled plant expressed by Laplace transfer function, its inversion will make the order of the numerator higher than the order of the denominator, which cannot be achieved in the actual control system. Therefore, this paper introduces UM-dynamic inversion algorithm to design the plant’s inversion part in UTDF-IMC structure.

From Figure 5, the system output $y=y_{m}+y_{e}$. Therefore,
(24)$$y=\left(rf-y_{e}F\right)P_{u}{}^{-1}P_{u}+y_{e}=rf+y_{e}\left(1-Ff\right)$$

### 3.3. UTDF-IMC Design Procedures

Figure 5 presents the U-model-based Two-Degree-of-Freedom IMC system framework, where f and F are the designed feedforward and feedback filters, respectively. P is the controlled plant or process, which is allowed to be linear or nonlinear. $P_{u}$ is U-model-based approximation to the controlled plant P. $P_{u}{}^{-1}$ is the inverter designed by the U-model-based root-solving algorithm. From (15), the parameters absorbed by $\alpha_{j}$ can be obtained from the output signal $y_{m}$ of the plant model $P_{u}\left(s\right)$ and controller output u. In general, similar to the classical IMC design, UTDF-IMC system design has the following two steps:
Assume the controlled plant or process P is stable and bounded, and its inverse $P^{-1}$ exists. Use U-model to describe P or convert the plant model $P_{0}$ into its U-realization $P_{u}$. The specific U-modeling process can refer to Section 2.1. In contrast to the classical IMC or classical TDF-IMC, U-realization of the original model $P_{0}$ can comfortably cover nonlinear dynamics, therefore, remove linearization restrictions.Design filters f and F according to system control performance requirements, then re-optimize the parameters of the filters according to the controller output limit. The feedforward filter determines the system’s set-point tracking ability (response time) while the feedback filter determines the system’s robustness. Because the control system performance is completely designed according to the two filters independently, designers can select the appropriate filters according to performance requirements, hardware limitations, controller output limitations, etc.

### 3.4. Property Analysis


Property 1 (Dual stability): Assume the plant model is perfectly matched ($P_{u}=P$) and system disturbance is absent d=0, then from Table 1, the closed-loop stability is characterized by the stability of the plant P($P^{-1}$) and the feedforward filter f. In this case, the system output signal will be: y=rf.Property 2 (Perfect control): Assume that the dynamic inverter $P_{u}{}^{-1}$ is satisfied with $P_{u}=P$ and P stable, then the closed-loop system is stable and perfectly controlled. In this case, the system output is $y=rf+\left(1-F\right)d$. The faster the response speed of feedback filter F, the better the system robustness.Property 3 (Zero offset): Assume that the steady-state gain of the controller equals to steady-state gain of the inverse model, and this closed-loop system is input-output stable with this controller, then offset free control is obtained asymptotically to step or ramp type inputs and disturbances.Property 4: Separability of designing the tracking filter and the robust filter: This is shown in the tables, which UTDF-IMC has no product of the two filters Ff.


Comparison with IMC and TDF-IMC, Table 1 and Table 2 list the three IMC types of control system configurations against disturbance and model mismatching, respectively. For UTDF-IMC the typical properties are analyzed below.

From Table 1, the factor associated with d is called the disturbance rejection designed. It is clear that this rejection part only depends on the feedforward filter f in IMC, depends on two filters F and f in TDF-IMC but only depends on the feedback filter F in UTDF-IMC structure. In case of model mismatch, it can also use the output error signal $y_{e}$ to analyze the system performance in Table 2:

From Table 2, regarding UTDF-IMC, the function associated with $y_{e}$ is robustness designed, where $y_{e}$ absorbs all whole modeling error and system disturbance; the function associated with signal r is for tracking designed. Obviously, when the controller equals to plant model inversion, all the tracking design only depends on the feedforward filter f and robustness designed is the same as previously discussed. In summary, compared with the classical IMC and TDF-IMC structure, the main differences of UTDF-IMC structure are as follows:
Classical TDF-IMC structure can make tracking ability and robustness be designed separately but not wholly independent due to the product of *Ff* in robustness specification. The UTDF-IMC overcomes this shortcoming without resorting to a more complex structure. Therefore, when the robustness performance of the system is determined, UTDF-IMC structure will have a faster response speed than the classical TDF-IMC structure.U-model is used to facilitate control system design, which can be easily to form an inversion of the plants to cancel both dynamic and nonlinearities. Accordingly, it converts the nonlinear control system into a linear model-based control with a nonlinear dynamic inverter.UM-dynamic inversion algorithm is used to design the inversion part in UTDF-IMC structure, which has a faster convergence speed and allows the inversion part exists alone properly without the feedforward filter.This structure where feedforward filter f from outside the control loop allows the tracking ability and robustness performance to be completely independently designed.The improved control performance is not complicating the system structure and/or increasing the additional computation burden throughout the design process.

## 4. Simulation Demonstrations

This simulation demonstration selects three plants to test the proposed U-model-based TDF-IMC structure. Both plants will be controlled by IMC, TDF-IMC, and UTDF-IMC structure.

### 4.1. Linear Internal Model (Also Called Nominal Model in the Study)

(25)$$P_{0}\left(s\right)=\frac{\omega_{n}{}^{2}}{s^{2}+2\zeta \omega_{n}s+\omega_{n}{}^{2}}=\frac{1}{s^{2}+3s+1}$$

This is characterized with the damping ratio ζ=1.5 and the undamped natural frequency $\omega_{n}=1$.

For designing the UTDF-IMC system:
Convert plant model (25) into its corresponding U-model:
(26)$$\left\{\begin{array}{c} P_{u}\left(s\right):\;\ddot{y}=u-3\dot{y}-y=\alpha_{0}f_{0}\left(u\right)+\alpha_{1}f_{1}\left(u\right)\\ \alpha_{0}=-3\ddot{y}\frac{1}{s}-\ddot{y}\frac{1}{s^{2}},\;f_{0}\left(u^{0}\right)=1\\ \alpha_{1}=1,\;f_{1}\left(u\right)=u \end{array}\right.$$Design the inverter of the plant model $P_{u}\left(s\right)$:(27)$$u=\ddot{y}+3\ddot{y}\frac{1}{s}+\ddot{y}\frac{1}{s^{2}}$$Design feedforward filter $f\left(s\right)$ and feedback filter $F\left(s\right)$

In this paper, based on the UTDF-IMC system design procedure in Section 3.2, to make the system achieve a fast response speed and no overshoot, $f\left(s\right)=\frac{1}{{\left(0.2s+1\right)}^{2}}$ and $F\left(s\right)=\frac{1}{{\left(0.1s+1\right)}^{2}}$. To compare control performance fairly, TDF-IMC system uses the same filters as UTDF-IMC. To ensure the same robustness, the classical IMC system uses $f^{\prime }\left(s\right)=\frac{1}{{\left(0.1s+1\right)}^{2}}$.

To test the performance of the designed control system, assume the plant a 2nd order dynamic with ζ=1 and $\omega_{n}=0.5$, and an external disturbance added at the system output, i.e.,
(28)$$P\left(s\right)=\frac{1}{4s^{2}+1s+1}+D\left(s\right)$$

The system disturbance is a band-limit white noise with changing rate of 1hz, system signal-noise ratio (SNR) of 26.9db. The noise sequence is shown in Figure 6.

Figure 7 shows the simulation results under the three different IMC schemes. From Figure 7a,b, UTDF-IMC and IMC have better robustness performance in rejection of system disturbance and modeling error. IMC system has a faster tracking speed because of its fast respond-speed filter; however due to modeling errors, stronger tracking ability brings larger overshoot. The simulation results also demonstrate the analysis in Section 3.4. From Figure 7c, UTDF-IMC structure does not increase the maximum peak output of the controller compared with TDF-IMC structure. However, fast tracking speed also brings a large controller output peak in the IMC system, which may cause the controller to overload in real-time applications. Consider the control performance and controller load, in case of selecting the same filters (control parameters), UTDF-IMC system shows better control performance.

### 4.2. Nonlinear Internal Model

(29)$$P_{0}:\;\dot{y}=a{\dot{u}}^{3}+b{\dot{u}}^{2}-c\dot{u}-ky+e^{u}$$
where the coefficients a=b=c=1, k=0.5, then $P_{0}\left(s\right)={\dot{u}}^{3}+{\dot{u}}^{2}-\dot{u}-0.5y+e^{u}$.

For designing the UTDF-IMC system,
Convert plant model (29) into its corresponding U-model:
(30)$$\left\{\begin{array}{c} P_{u}\left(s\right):\dot{y}=\alpha_{0}f_{0}\left(\dot{u}\right)+\alpha_{1}f_{1}\left(\dot{u}\right)+\alpha_{2}f_{2}\left(\dot{u}\right)+\alpha_{3}f_{3}\left(\dot{u}\right)\\ \alpha_{0}=-0.5\dot{y}\frac{1}{s}+e^{u},\;f_{0}\left({\dot{u}}^{0}\right)=1\\ \alpha_{1}=1,\;f_{1}\left(\dot{u}\right)=-\dot{u}\\ \alpha_{2}=1,\;f_{1}\left({\dot{u}}^{2}\right)={\dot{u}}^{2}\\ \alpha_{3}=1,\;f_{1}\left({\dot{u}}^{3}\right)={\dot{u}}^{3} \end{array}\right.$$Design the inverter of the plant model $P_{u}\left(s\right)$:
(31)$$u=root\left(\alpha_{0}f_{0}\left(\dot{u}\right)+\alpha_{1}f_{1}\left(\dot{u}\right)+\alpha_{2}f_{2}\left(\dot{u}\right)+\alpha_{3}f_{3}\left(\dot{u}\right)-\dot{y}=0\right)$$

It should be noted that because Equation (32) is a cubic equation of one variable about $\dot{u}$, to ensure that the controller output is rational, the real root of Equation (32) is selected as the output of the controller.

3.Design feedforward filter $f\left(s\right)$ and feedback filter $F\left(s\right)$

Same as previous work, to make the system achieve a fast response speed and no overshoot, this paper chooses $f\left(s\right)=\frac{1}{{\left(0.1s+1\right)}^{2}}$ and $F\left(s\right)=\frac{1}{{\left(0.2s+1\right)}^{2}}$ for the plant 2. To compare control performance fairly, TDF-IMC system uses the same filters as UTDF-IMC. To ensure the same tracking speed, the classical IMC system uses $f^{\prime }\left(s\right)=\frac{1}{{\left(0.2s+1\right)}^{2}}$.

To demonstrate the performance of the designed control system, assume plant with the same structure as the IM, but c=1.4 and k=0.8, and an external noise added at the system output, i.e.,
(32)$$P\left(s\right):\;\dot{y}={\dot{u}}^{3}+1.4{\dot{u}}^{2}-\dot{u}-0.8y+e^{u}+d$$

The system noise is a band-limit white noise with changing rate of 1 hz and SNR of 20.9 db. The noise sequence is shown in Figure 8.

Figure 9a–c show the simulation results under the three IMC schemes, Figure 9d shows the tracking reference signal. From Figure 9a,b, UTDF-IMC both has a better robustness performance in rejection of system disturbance and modeling error and faster tracking speed. When the reference signal suddenly jumps sharply, the response of TDF-IMC system also shakes sharply although it has the same filters as UTDF-IMC’s. These simulation results demonstrate the analysis in Section 3.4. From Figure 9c, UTDF-IMC structure does not increase the burden on the controller, although it has a better control performance. The outputs of controller show that the UTDF-IMC is not overloaded. Once again, consider the control performance and controller load, in case of selecting the same filters (control parameters), UTDF-IMC system shows better control performance.

### 4.3. Control of PMSM System

In the past few decades, Permanent Magnet Synchronous Motors (PMSM) have been widely used in industry because of their high-power density, high efficiency, and large torque inertia ratio. PMSM is essentially a nonlinear Multiple-Input-Multiple-Output (MIMO) system, so parameter uncertainty and interference acting on torque will make it difficult for PMSM control systems to obtain higher control performance [36]. Most advanced control strategies [37,38,39] for PMSM servo system position control ignore the nonlinear term in the speed equation, assuming that A=B and load torque disturbance does not change. Therefore, it is still a challenge to provide an efficient set-point value tracking control strategy for a general PMSM system affected by time-varying system disturbance and uncertain parameters. Therefore, this section applies the proposed UTDF-IMC structure combined with the U-modeling of the PMSM system to achieve high-precision set-point robust tracking control of the PMSM operation.

#### 4.3.1. Modeling of PMSM System

It should be noted that the permanent magnets used in the PMSM are a type of modern rare-earth varieties with high resistivity, so the induced current in the rotor can be negligible. The model of the PMSM is based on s number of equations in the d-q reference frame [40].

The electric torque of the PMSM is:(33)$$T_{e}=3p\left[\Phi_{v}i_{q}+\left(L_{d}-L_{q}\right)i_{d}i_{q}\right]/2$$

And its motor dynamics can be modeling as:(34)$$T_{e}=T_{L}+B\omega_{r}+J\Delta \omega_{r}$$

The relationship between voltages and currents in motor are:(35)$$\left[\begin{array}{c} V_{d}\\ V_{q} \end{array}\right]=\left[\begin{array}{cc} R_{s}+L_{d}\Delta & -p\omega_{r}L_{q}\\ p\omega_{r}L_{d} & R_{s}+L_{q}\Delta \end{array}\right]\left[\begin{array}{c} i_{d}\\ i_{q} \end{array}\right]+\left[\begin{array}{c} 0\\ p\omega_{r}\Phi_{v} \end{array}\right]$$

The rotor flux rotates at rotor speed $\omega_{r}$ and is positioned by the rotor angular position:(36)$$\theta_{r}=\int \omega_{r}dt$$

Therefore, the PMSM in the rotating *d-q* reference frame can be modeled in the following state-space equation [41],
(37)$$\left\{\begin{array}{c} \frac{d\theta_{r}}{dt}=\omega_{r}\\ \frac{d\omega_{r}}{dt}=\frac{3p\Phi_{v}}{2J}i_{q}+\frac{3p}{2J}\left(L_{d}-L_{q}\right)i_{d}i_{q}-\frac{B}{J}\omega_{r}-\frac{1}{J}T_{L}\\ \frac{di_{d}}{dt}=-\frac{R_{s}}{L_{d}}i_{d}+\frac{pL_{q}}{L_{d}}i_{q}\omega_{r}+\frac{1}{L_{d}}V_{d}\\ \frac{di_{q}}{dt}=-\frac{R_{s}}{L_{d}}i_{q}-\frac{pL_{d}}{L_{q}}i_{d}\omega_{r}-\frac{p\Phi_{v}}{L_{q}}\omega_{r}+\frac{1}{L_{q}}V_{q} \end{array}\right.$$
where

Δ: differential operator ($\frac{d*}{dt}$)

$\theta_{r}$ and $\omega_{r}$: the rotor angular position and rotor speed

$i_{d}$, $i_{q}$ and $V_{d}$, $V_{q}$: stator currents and voltages in *d-q* reference frame

$L_{d}$ and $L_{q}:$ axes inductances in *d-q* reference frame

$T_{L}$: load torque, $\Phi_{v}$:rotor flux, J: inertia, $R_{s}$: stator resistance, B: viscous friction coefficient and p: number of pole pairs.

The design aim is controlling voltages $V_{d}$ and $V_{q}$ in (37) to make rotor position $\theta_{r}$ track a desired constant reference position $\theta_{d}$ and the current $i_{d}$ is regulated to zero asymptotically, concretely, this PMSM control system is two-input two-output with $u=\left[u_{1}\;u_{2}\right]=\left[V_{d}\;V_{q}\right]$ and $y=\left[y_{1}\;y_{2}\right]=\left[\theta_{r}\;i_{d}\right]$. The same as used [41], the commonly used nonlinear load torque disturbance to test the system performance is generated by the following disturbance dynamic model:(38)$$\left\{\begin{array}{c} {\dot{v}}_{1}=v_{2}\\ {\dot{v}}_{2}=-av_{1}+b\left(1-v_{1}{}^{2}\right)v_{2} \end{array}\right.$$
where $v_{1}=T_{L}$ is the solution of this Van der Pol oscillator.

Let
(39)$$\left\{\begin{array}{c} x_{1}=\theta_{r},x_{2}=\omega_{r},x_{3}=i_{d},x_{4}=i_{q}\\ a_{1}=\frac{3p\Phi_{v}}{2J},a_{2}=\frac{3p}{2J}\left(L_{d}-L_{q}\right),a_{3}=\frac{B}{J},a_{4}=\frac{1}{J}\\ b_{1}=\frac{R_{s}}{L_{d}}i_{d},b_{2}=\frac{pL_{q}}{L_{d}},b_{3}=\frac{1}{L_{d}}\\ c_{1}=\frac{R_{s}}{L_{d}},c_{2}=\frac{pL_{d}}{L_{q}},c_{3}=\frac{p\Phi_{v}}{L_{q}},c_{4}=\frac{1}{L_{q}} \end{array}\right.$$

Then system (37) can be rewritten into standard state-space equation of:(40)$$\left\{\begin{array}{c} {\dot{x}}_{1}=x_{2}\\ {\dot{x}}_{2}=a_{1}x_{4}+a_{2}x_{3}x_{4}-a_{3}x_{2}-a_{4}v_{1}\\ {\dot{x}}_{3}=-b_{1}x_{3}+b_{2}x_{3}x_{2}+b_{3}u_{1}\\ {\dot{x}}_{4}=-c_{1}x_{4}-c_{2}x_{3}x_{2}-c_{3}x_{2}+c_{4}u_{2} \end{array}\right.\;\mathrm{and}\;\left\{\begin{array}{c} y_{1}=x_{1}+d\\ y_{2}=x_{3} \end{array}\right.$$
where d is the system disturbance. Linearize system (40) gives:(41)$$\dot{\bar{x}}=A\bar{x}+B\bar{u},\;\bar{y}=C\bar{x}+d$$
where $A=\left[\begin{array}{cc} \begin{array}{c} \begin{array}{c} 0\\ 0 \end{array}\\ 0\\ 0 \end{array} & \begin{array}{ccc} \begin{array}{c} \begin{array}{c} 1\\ -a_{3} \end{array}\\ -b_{3}\\ 0 \end{array} & \begin{array}{c} \begin{array}{c} 0\\ a_{1} \end{array}\\ -b_{1}\\ 0 \end{array} & \begin{array}{c} \begin{array}{c} 0\\ 0 \end{array}\\ 0\\ -c_{1} \end{array} \end{array} \end{array}\right]$, $B=\left[\begin{array}{cc} 0 & 0\\ 0 & 0\\ \begin{array}{c} b_{4}\\ 0 \end{array} & \begin{array}{c} 0\\ c_{3} \end{array} \end{array}\right]$ and $C=\left[\begin{array}{ccc} 1 & 0 & \begin{array}{cc} 0 & 0 \end{array}\\ 0 & 0 & \begin{array}{cc} 1 & 0 \end{array} \end{array}\right]$.

#### 4.3.2. Simulation Test

In this section, the following three controllers are compared with simulation tests.

IMC: The filter time parameter shown in equation (9) is chosen as λ=0.01, use linearization to approximate the inverse of PMSM.TDF-IMC: Based on the structure in Figure 4, the feedforward filter and feedback filter are chosen as $f=\frac{1}{{\left(1+0.1s\right)}^{\gamma }},\;F=\frac{1}{{\left(1+0.01s\right)}^{\gamma }}$, use UM-dynamic inversion to design the inverse of PMSM.UTPF-IMC: To test the performance of UTDF-IMC fairly, based on the structure in Figure 5, the feed forward filter and feedback filter are chosen as $f=\frac{1}{{\left(1+0.1s\right)}^{\gamma }},\;F=\frac{1}{{\left(1+0.01s\right)}^{\gamma }}$, use UM-dynamic inversion to design the inverse of PMSM.

Comparison test of controller a and controller c is to demonstrate the superiority of UM-dynamic inversion algorithm for modeling nonlinear controlled plants/processed and inversion calculation, and comparison test of controller b and controller c is to show the efficiency of the proposed UTPF-IMC structure under the same modeling and calculation accuracy. The nominal values of PMSM parameters [41] for the simulations are p=3, $R_{s}=1.2\;\Omega$, $\Phi_{v}=0.18\;\mathrm{Vs}/\mathrm{rad}$, $\bar{L_{d}}=0.011\;H$, $\bar{L_{q}}=0.015\;H$, $\bar{B}=0.0001\;\mathrm{Nms}/\mathrm{rad}$
$\bar{J}=0.006\;{\mathrm{kgm}}^{2}$. Choose a=9, b=1. The initial values are chosen as follows: $\theta_{r}\left(0\right)=0\;\mathrm{rad},\;\omega_{r}\left(0\right)=0\;\mathrm{rad}/s,\;i_{d}\left(0\right)=0\;A,\;i_{q}\left(0\right)=0\;A.$

#### 4.3.3. Matched Model with System Disturbance

To test the property 2 in Section 3.4 while the process model is perfectly matched, i.e., $P_{0}=P$, assign the step reference signal with tracking positions $\theta_{d}=\pi \;\mathrm{rad}$ and current $i_{d}=0$, plus a squared disturbance shown in Figure 10 is added.

Figure 11 shows the simulation results. Clearly, all the controllers can track the desired set-point and reject the system disturbance but the robustness of TDF-IMC system is worse than others. IMC system has a faster response speed; however, it has overshoot due to linearization error. From Figure 11b, when $\theta_{r}$ reaches the designated angular position, rotor speed $\omega_{r}$ is stabilized at zero. From Figure 11c, all control systems current $i_{d}$ can stay at 0, but its peak value in IMC system is much larger than the others obviously. These simulation results demonstrate properties justified in Section 3.4. From Figure 11d,e, the controller outputs have large peak values at initial phase in the IMC system, especially output voltage $V_{d}$.

#### 4.3.4. Mismatched Model with System Disturbance

In this part, three controllers under a more actual situation (with modeling error) will be tested to investigate property 3 in Section 3.4. The parameters of PMSM become: $L_{d}=0.5\bar{L_{d}},\;L_{q}=1.3\bar{L_{q}},\;B=1.45\bar{B},\;J=0.75\bar{J}.$ The load torque disturbance generated by (38) with initial values of $v_{1}\left(0\right)=0$ and $v_{2}\left(0\right)=0.1$ is also added in PMSM system, which is shown in Figure 12. System disturbance is the same as previous experiment shown in Figure 10a and Figure 13 shows the comparative simulation results.

From Figure 13a, IMC system has tracking error due to the accuracy limitation of linearization, which makes IMC unable to reject strong nonlinear load torque disturbance. Both TDF-IMC and UTDF-IMC systems can achieve the prescribed set-point tracking performance because UM-dynamic inversion does not lose any nonlinear features. UTDF-IMC system has better robustness than TDF-IMC system due to the difference in their structures. From Figure 13b, when $\theta_{r}$ reaches the designated angular position, rotor speed $\omega_{r}$ in UTDF-IMC and TDF-IMC systems is stabilized at zero; however, the rotor revolves slightly in IMC system. From Figure 13c, all current $i_{d}$ staying at zero but its peak value with IMC is larger than the others, this is because of the cost of faster response speed in IMC system. From Figure 13d,e, the controller outputs also have large peak values at initial phase in the IMC system, especially output voltage $V_{d}$.

In summary, from all simulation results, the control system using the linearization method does degrade the control performance while there is a strong nonlinear disturbance. Additionally, by using UM-dynamic inversion, UTDF-IMC and TDF-IMC systems can achieve reasonably good set-point tracking performance, and UTDF-IMC system has better robustness than classical TDF-IMC system with the same parameters chosen in the filters.

## 5. Conclusions

This paper introduces an effective U-model-based Two-Degree-of-Freedom IMC framework. Consistently with the simulation test results of linear and nonlinear controlled plants, the proposed UTDF-IMC framework shows its strong robustness and effectiveness in control system design compared with the classical IMC and TDF-IMC approaches. It is believed that UTDF-IMC, enhanced with nonlinear dynamic inverter, could be applied more effectively to a wide range of industrial control system design. Therefore, this study has established a platform for possible further expansion, for example controlling Multi-Input and Multi-Output (MIMO) systems, which involves solution challenges with nonlinear set equation in case of under, full, and over actuated control system design. Another research direction is to expand the UTDF-IMC to deal with nonminimum phase/unstable zero dynamic systems.

## Figures and Tables

**Figure 1 entropy-23-00169-f001:**
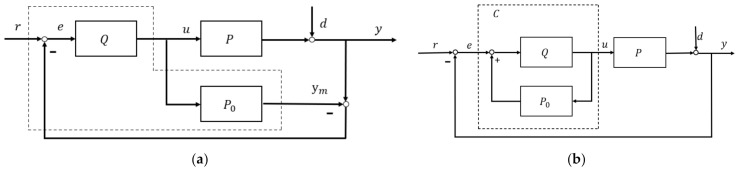
IMC method structure (**a**) Internal model control structure; (**b**) Equivalent IMC structure.

**Figure 2 entropy-23-00169-f002:**
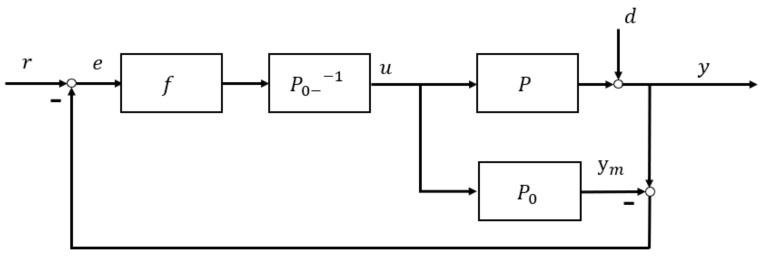
Robust IMC structure.

**Figure 3 entropy-23-00169-f003:**
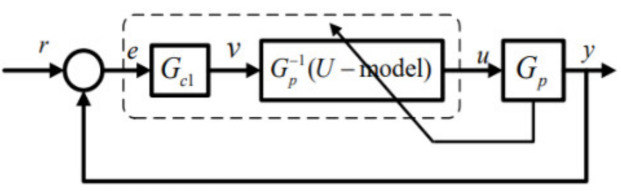
U-control system framework.

**Figure 4 entropy-23-00169-f004:**
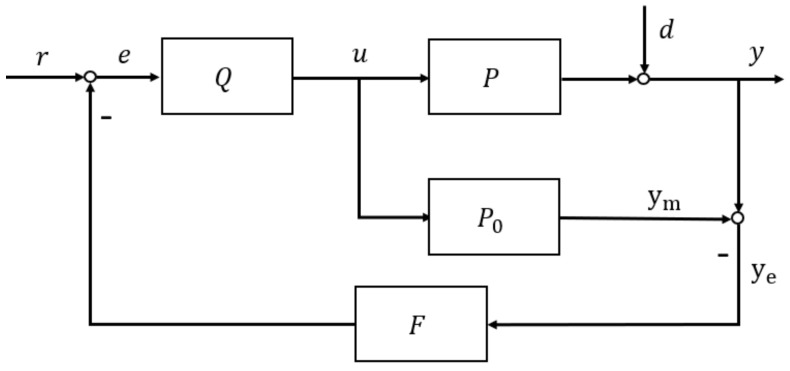
IMC structure with the feedback filter.

**Figure 5 entropy-23-00169-f005:**
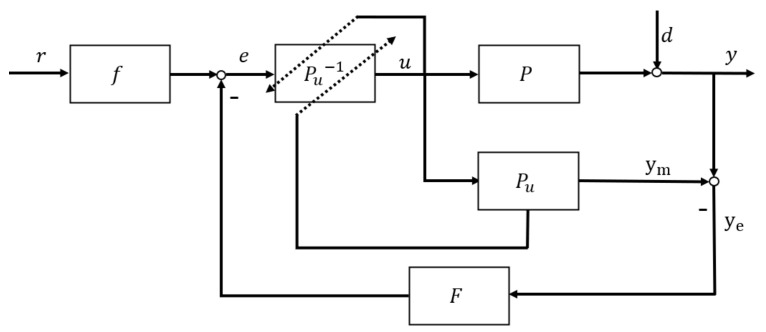
U-model-based Two-Degree-of-Freedom IMC structure.

**Figure 6 entropy-23-00169-f006:**
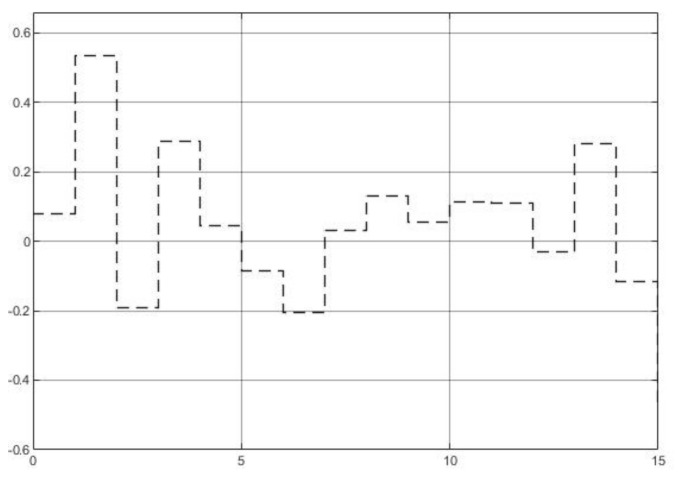
System disturbance noise.

**Figure 7 entropy-23-00169-f007:**
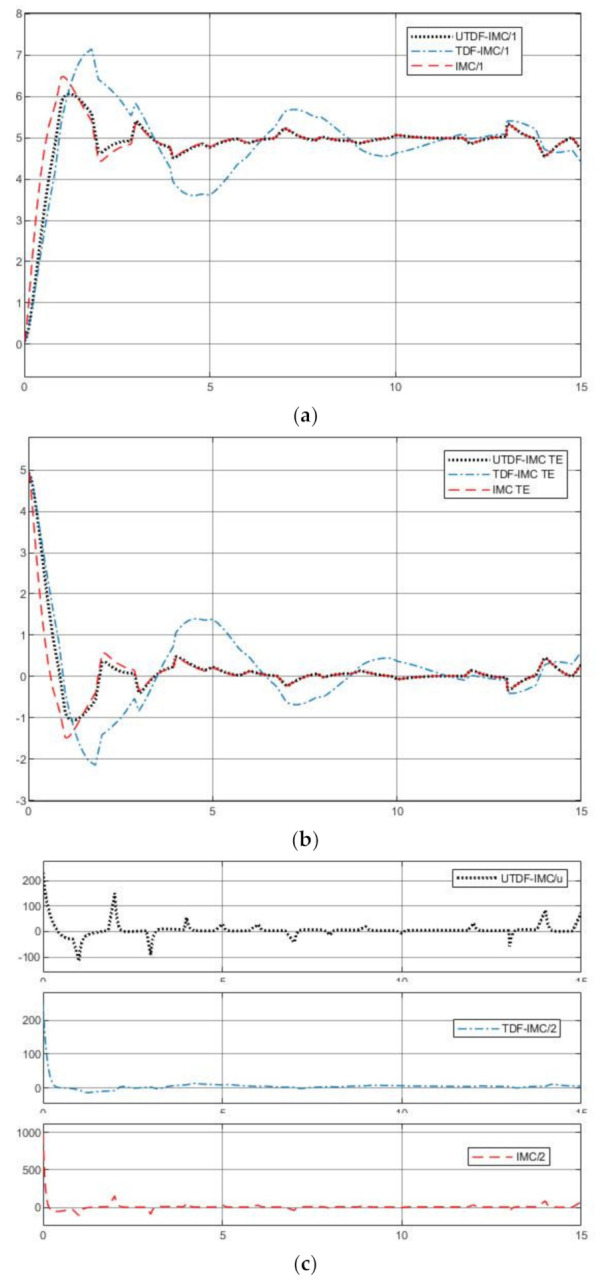
Simulation results of plant1 (**a**) System outputs; (**b**) Tracking errors; (**c**) Controller outputs.

**Figure 8 entropy-23-00169-f008:**
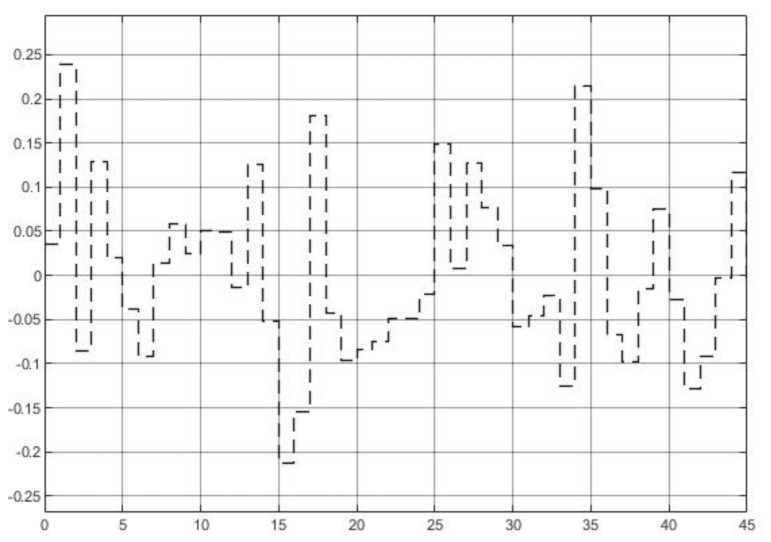
System noise.

**Figure 9 entropy-23-00169-f009:**
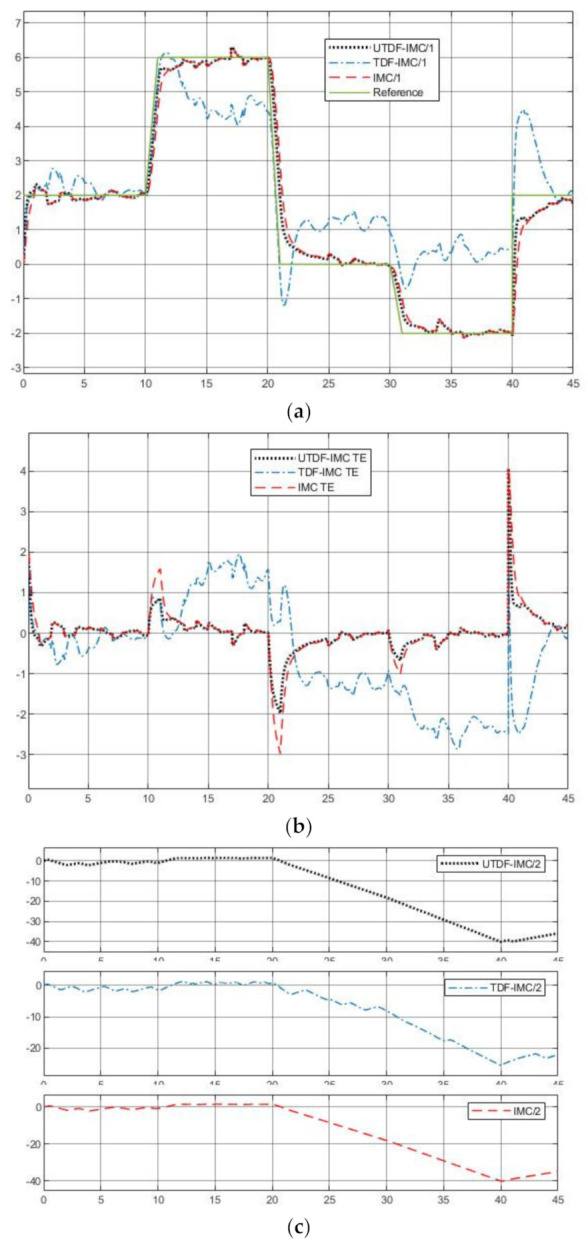
Simulation results of plant2 (**a**) System outputs; (**b**) Tracking errors; (**c**) Controller outputs; (**d**) Tracking reference.

**Figure 10 entropy-23-00169-f010:**
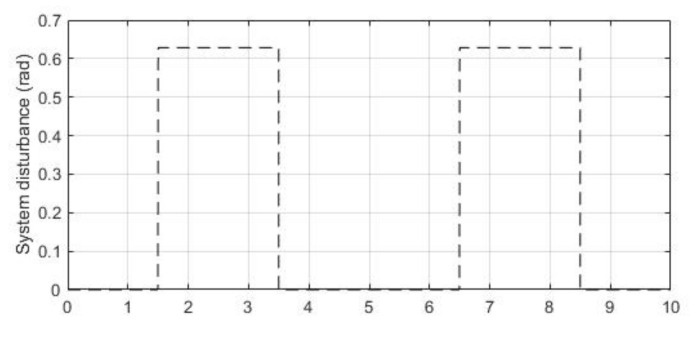
System disturbance.

**Figure 11 entropy-23-00169-f011:**
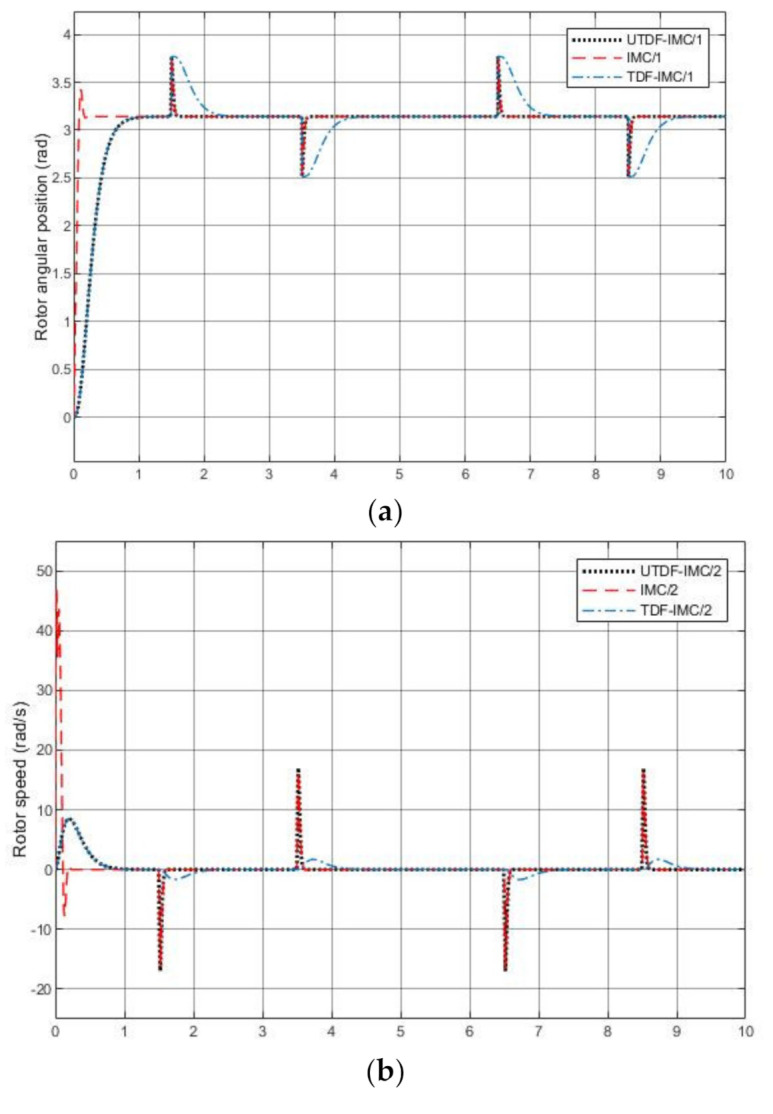
Simulation results with only system disturbance (**a**) Output angular position $\theta_{r}$; (**b**) Output rotor speed $\omega_{r}$; (**c**) Output current $i_{d}$; (**d**) Controller output voltage $V_{d}$; (**e**) Controller output voltage $V_{q}$.

**Figure 12 entropy-23-00169-f012:**
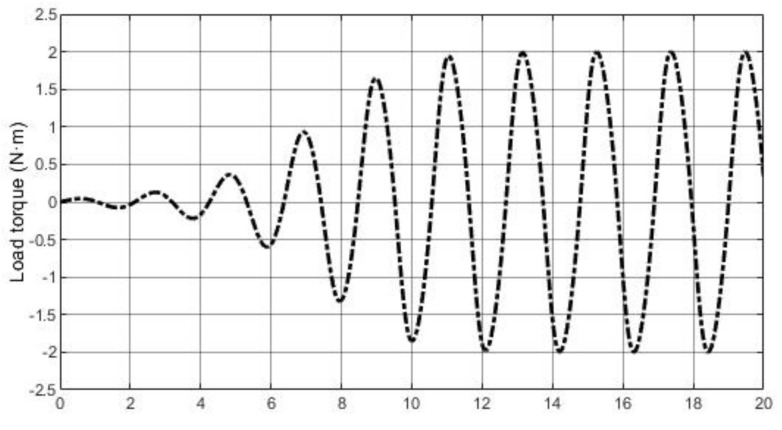
Load torque disturbance.

**Figure 13 entropy-23-00169-f013:**
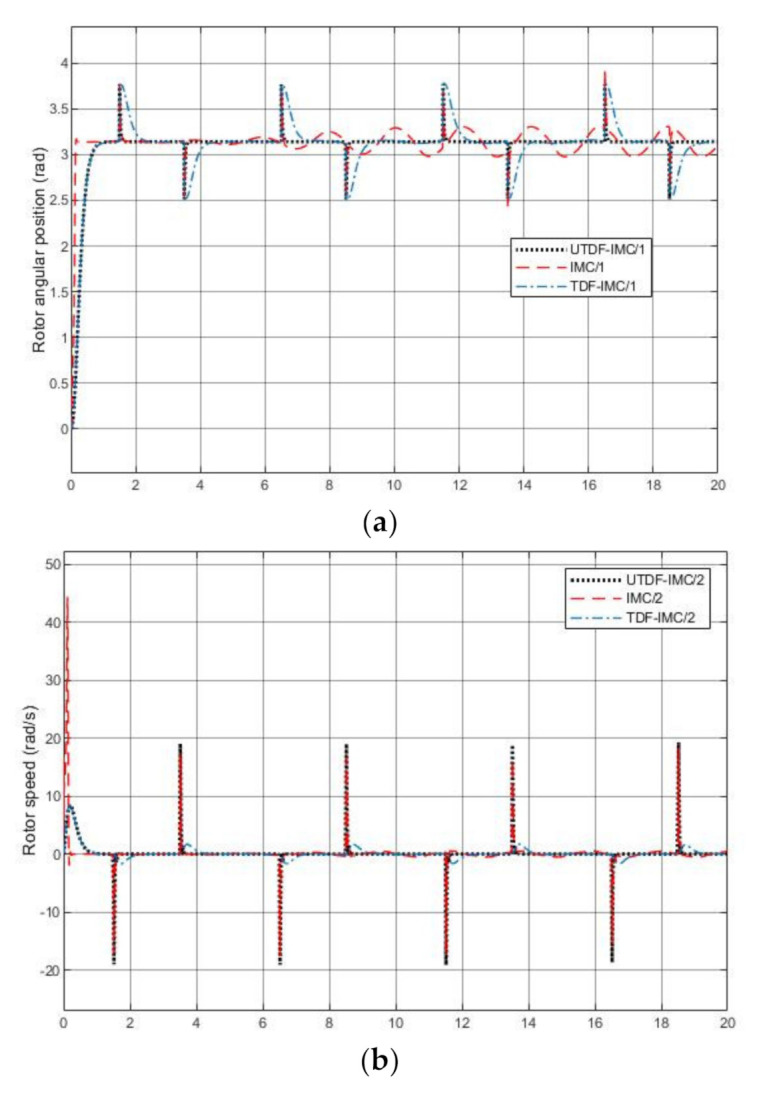
Simulation results with modeling error and disturbance (**a**) Output angular position $\theta_{r}$; (**b**) Output rotor speed $\omega_{r}$; (**c**) Output current $i_{d}$; (**d**) Controller output voltage $V_{d}$; (**e**) Controller output voltage $V_{q}$.

**Table 1 entropy-23-00169-t001:** Input/output comparison of IMC, TDF-IMC, and UTDF-IMC against disturbance.

	Controller Output u	System Output y
IMC	$u=\frac{1}{P_{0}+\left(P-P_{0}\right)f}\left(rf-df\right)$	$y=\frac{fP}{P_{0}+\left(P-P_{0}\right)f}r+\frac{P_{0}\left(1-f\right)}{P_{0}+f\left(P-P_{0}\right)}d$
TDF-IMC	$u=\frac{1}{P_{0}+\left(P-P_{0}\right)Ff}\left(rf-dFf\right)$	$y=\frac{fP}{P_{0}+\left(P-P_{0}\right)Ff}r+\frac{P_{0}\left(1-fF\right)}{P_{0}+\left(P-P_{0}\right)Ff}d$
UTDF-IMC	$u=\frac{1}{P_{u}+\left(P-P_{u}\right)F}\left(rf-dF\right)$	$y=\frac{fP}{P_{u}+\left(P-P_{u}\right)F}r+\frac{P_{u}\left(1-F\right)}{P_{u}+\left(P-P_{u}\right)F}d$
	where $P_{u}$ is the U-realization of $P_{0}$	

**Table 2 entropy-23-00169-t002:** Output comparison of IMC, TDF-IMC, and UTDF-IMC against model mismatching.

	System Output y
IMC	$y=rf-\left(1-f\right)y_{e}$
TDF-IMC	$y=rf-\left(1-Ff\right)y_{e}$
UTDF-IMC	$y=rf+\left(1-F\right)y_{e}$

## Data Availability

The data used to support the findings of this study are available from the corresponding author upon request.

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
