# Peer review of "U-Model-Based Two-Degree-of-Freedom Internal Model Control of Nonlinear Dynamic Systems"

_entropy, 2021, doi:10.3390/e23020169_

Round 1
Reviewer 1 Report
The paper is somewhat interesting in the field of internal model control of nonlinear dynamic systems. However, it can be improved by making some additions and corrections such as:
(1) The sentence is incorrect expressed:
Where the controller is:
?=?/(1−??0 ) (2.1)
(2) What is ym in the sentence below?
The control system is designed to keep the output ? following a set-point value ? in a pre-specified ?? with desired transient and steady state performance.
(3) The sentence is incorrect expressed:
UTDF-IMC and TDF-IMC systems can achieve perfect set-point tracking performance, and UTDF-IMC system has better robustness than classical TDF-IMC system with the same parameters chosen in filters.
Actually, there is no perfect set-point tracking performance.
(4) Improving the English language of the paper. Here are some examples of unclear and wrong sentences:
- The IMC structure exhibits robustness to overcome model uncertainties and system disturbances, which the systems are effectively systematically to design and tune, and therefore successfully applied in different industries [3-5].
- The IMC structure exhibits robustness to overcome model uncertainties and system disturbances, which the systems are effectively systematically to design and tune, and therefore successfully applied in different industries [3-5].
- In almost all real industrial process control, overcoming disturbances is the main task of the control system, and model uncertainty is inevitable. Consequently, the feedback error ? in action to form a closed-loop control structure.
- Because ? and ?0 generally do not match.
- The time-varying parameter ??∈ℝ+ absorbing all other output terms like [(?−1)?,…,?]∈ℝ? and input terms [(?−1)?,…,?]∈ℝ?.
- The UM-dynamic inversion algorithm, that is the solution of ??−1, is to obtain the input ? by solving the roots from (2.11), that is,
An algorithm is not a solution. Solving equation (2.11), not the roots.
(5) Insertion of some recent references from the last three years in MDPI Journals (for example: Cirtoaje V. A Practical Unified Algorithm of P-IMC Type, Processes, 2020, 8(2) , 165).
Author Response
Reviewer 1
The paper is somewhat interesting in the field of internal model control of nonlinear dynamic systems. However, it can be improved by making some additions and corrections such as:
(1) The sentence is incorrect expressed:
Where the controller is:
?=?/(1−??0 ) (2.1)
*** Re-phrased as < Figure 1b shows the equivalently rearranged IMC structure, which the controller is expressed in the inner loop >
?=?/(1−??0 ) (2.1)
(2) What is ym in the sentence below?
The control system is designed to keep the output ? following a set-point value ? in a pre-specified ?? with desired transient and steady state performance.
*** Revised as < For a given set-point reference r, the control system is designed to keep the output y following a pre-specified output response ym with the desired transient and steady state performance >
(3) The sentence is incorrect expressed:
UTDF-IMC and TDF-IMC systems can achieve perfect set-point tracking performance, and UTDF-IMC system has better robustness than classical TDF-IMC system with the same parameters chosen in filters.
Actually, there is no perfect set-point tracking performance.
*** PERFECT CONTROL is one of the important ICM notions/properties, which the mentioned control systems are satisfied. Sorry for mixing up in the sentence, it should be <*** reasonably good *** >
(4) Improving the English language of the paper. Here are some examples of unclear and wrong sentences:
*** Yes, we have done our best to fine tune the draft throughout. Please check the revised.
The IMC structure exhibits robustness to overcome model uncertainties and system disturbances, which the systems are effectively systematically to design and tune, and therefore successfully applied in different industries [3-5].
*** Revised as <The IMC structure is characterised with 1) capable robustness to overcome model uncertainties and system disturbances, 2) effective procedures for designing and tuning, 3) successful application across different industries [3-5]>
In almost all real industrial process control, overcoming disturbances is the main task of the control system, and model uncertainty is inevitable. Consequently, the feedback error ? in action to form a closed-loop control structure.
*** As requested from the second reviewer, we have tailored the basic IMC introduction for whom are familiar with the subject. These sentences are removed in the revised draft accordingly.
Because ? and ?0 generally do not match.
*** As requested from the second reviewer, we have tailored the basic IMC introduction for whom are familiar with the subject. This sentence is removed in the revised draft accordingly.
The time-varying parameter ??∈ℝ+ absorbing all other output terms like [(?−1)?,…,?]∈ℝ? and input terms [(?−1)?,…,?]∈ℝ?.
*** Revised as <The time-varying parameter ??∈ℝ+ is a function of the other outputs [(?−1)?,…,?]∈ℝ? and inputs [(?−1)?,…,?]∈ℝ?. >
The UM-dynamic inversion algorithm, that is the solution of ??−1, is to obtain the input ? by solving the roots from (2.11), that is,
An algorithm is not a solution. Solving equation (2.11), not the roots.
*** revised as < For determining the output ? of ??−1, a general UM-dynamic inversion algorithm has been developed [19] to determine one of the solutions of from solving the following general polynomial equation>
(5) Insertion of some recent references from the last three years in MDPI Journals (for example Cirtoaje V. A Practical Unified Algorithm of P-IMC Type, Processes, 2020, 8(2) , 165).
*** Yes, we have quoted the following references with MDPI publications
Kawan, C. 2019, Special Issue, Entropy in networked control, Entropy. https://www.mdpi.com/journal/entropy/special_issues/control
Cirtoaje V. A Practical Unified Algorithm of P-IMC Type, Processes, 2020, 8(2), 165
*** In addition, a paragraph has been supplemented to link <entropy> to digital control systems in the revised introduction.
The other practical topic is digital control, which has been used in almost all modern engineering (manmade) systems. To deal with a digital channel between the sensor and the controller, entropy [40], the concepts, terminologies, and techniques has been adopted in digital control systems. The invariance entropy [41] has been used to determine the smallest average rate of information transmission to guarantee a considered subset of the state space invariant to achieve the integrated control system performance. Accordingly control system design in connection to online applications, with a potential contribution to this field, is probably making the controllers more efficient in online computation, so that reduces the burden to the communication capacity or at least does not increase significantly the information processing complexity.
*** Invariance entropy is added in Keywords
*** <The authors are grateful to the editors and the anonymous reviewers for their constructive comments and suggestions with regard to the revision of the paper> has been added in the Acknowledgements.

Reviewer 2 Report
The paper focuses in a U-model based two-degree-of-freedom internal model control (UTDFIMC) structure with strength in nonlinear dynamic inversion, and separation of tracking design and robustness. However, there are some issues to be addressed as follows:
- The problem statement should be better introduced.
- The contributions can be better introduced in light of existing literature.
- The simulation results section is very weak and for example does not consider the robustness of the control developed. Moreover, the physical magnitudes of the axes of the simulation results are not indicated
Authors must be noted that the reader of their paper is absolutely familiar with the subject IMC. Therefore, there is no need to explain the subject IMC from the basics as much as authors did in this paper.
The authors should polish the paper suitably. The whole paper should be reviewed carefully, in order to correct all the errors. Some grammar issues and typos need to be corrected.
References are not in the format established by the journal
Author Response
Reviewer 2
The paper focuses in a U-model based two-degree-of-freedom internal model control (UTDFIMC) structure with strength in nonlinear dynamic inversion, and separation of tracking design and robustness. However, there are some issues to be addressed as follows:
- The problem statement should be better introduced.
*** Yes, we have fine-tuned the introduction with the following structures in light of having critically reviewed the literatures
- Commonly concerned foundation with PID, model, control, tuning, uncertainty, robustness, entropy in control system design (progression, topics for attention)
- IMC, linear/nonlinear, nonlinear dynamic inversion, separation of filter designs (progression, bottleneck issues and justifications for further study)
- U-control (progression, potential solution platform for the new development)
- Contribution of the study
Should the reviewer feel to change the structure, we would like to follow.
- The contributions can be better introduced in light of existing literature.
*** Yes, the justification is based on current IMC lack of general nonlinear dynamic inversion (cancellation of both nonlinearities and dynamics) and proper separation of designing the two filters, which are critically reviewed in light of the existing representative literatures.
- The simulation results section is very weak and for example, does not consider the robustness of the control developed. Moreover, the physical magnitudes of the axes of the simulation results are not indicated
*** We have checked throughout all the simulations, which have included various robustness tests of the novel developed control and comparisons with two exiting representative control approaches. For further robustness study and computational experimental validation, we will present more in a future study of < L1 robust adaptive UIMC for non-affine nonlinear dynamic systems>. The other reason for not adding extra simulation plots is, even after tailoring the basic IMC introduction, the paper still takes up 30 pages in the revised version.
*** The physical magnitudes of the axes have been supplemented in the PMSM industrial system simulation plots.
Authors must be noted that the reader of their paper is absolutely familiar with the subject IMC. Therefore, there is no need to explain the subject IMC from the basics as much as authors did in this paper.
*** Yes, Section 2.1 IMC in the preliminary has been tailored for whom are familiar with the subject.
The authors should polish the paper suitably. The whole paper should be reviewed carefully, in order to correct all the errors. Some grammar issues and typos need to be corrected.
*** Yes, we have done our best to fine tune the draft throughout. Please check the revised.
References are not in the format established by the journal
*** From the journal website advice, this will be edited by the publisher once the paper accepted for publication.
